

# Observational study of factors influencing the dispersion of warm fog droplet spectrum in Xishuangbanna, China

Zhenya An[1,2], Xiaoli Liu[1,2*]

[1]China Meteorological Administration Aerosol-Cloud and Precipitation Key Laboratory, Nanjing University of Information
Science and Technology, Nanjing 210044, China.
[2]College of Atmospheric Physics, Nanjing University of Information Science and Technology, Nanjing 210044, China

*Correspondence to*: *Xiaoli Liu (liuxiaoli2004y@nuist.edu.cn)*

**Abstract.** The microphysical characteristics of fog and stratiform clouds are somewhat similar. The study of the microphysical characteristics of warm fog, fog droplet spectral relative dispersion, and their influencing factors can deepen our understanding of the variability and influencing factors of cloud droplet spectral relative dispersion, while also investigating the formation and maintenance mechanisms of fog. This is currently a scientific issue that still remains controversial in cloud physics research and climate prediction. In this paper, we analyzed three months of Xishuangbanna radiation fog observations to explore the microphysical characteristics of fog. The results show followings: (1) When the autoconversion threshold (T) increased to greater than 0.4, the positive correlation between the relative dispersion of fog droplet spectrum and the volume mean diameter or water content of fog droplet weakened, also the positive correlation between relative dispersion and number concentration increased where the main mechanism needed to be integrated considering the interaction of collision-coalescence, condensation, and nucleation processes. It is found that the strength of the collision-coalescence process has a certain influence on the variation rule of dispersion. (2) The number concentration of 2-12 µm droplets in the fog constrained the relationship between the T and relative dispersion, with the number of large droplets reflects the strength of the collision-coalescence process. (3) Supersaturation changed microphysical quantities by increasing the number concentration of small droplets in the fog, which affected the variations of relative dispersion. For supersaturation greater than 0.12 %, the number concentration of droplets larger than 30 µm may be decreased due to gravitational settling. In addition, there is no significant relationship between supersaturation and relative dispersion if the initial nucleated fog droplet spectrum is narrow .

## 1 Introduction

Fog is a common weather phenomenon where water vapor condenses (or freezes) and suspends in the atmospheric boundary layer near the Earth's surface, causing visibility to be less than 1 km. The existence of fog can easily lead to road traffic accidents. Although drivers face meteorological disasters such as ice, snow, wind, and fog on the road, ice, snow, and wind can be accurately predicted and reduced in harm through sanding, snow removal machines, as well as designing road sections and bridge structures (Musk, 1991). However, due to the complex formation mechanism and variable atmospheric background, it is very difficult to forecast the intensity of fog. The economic losses related to fog and low visibility are even comparable to



those caused by other weather events such as tornadoes and hurricanes (Gultepe et al., 2007). Therefore, the impact of fog on transportation has gradually been given more attention. In addition, the temperature inversion structure during fog formation is also unfavorable for the diffusion of pollutants, seriously affecting air quality. Fog itself also has a certain self-purification effect on the atmosphere, but when the concentration of pollutants in the atmosphere exceeds the self-purification capacity of
the atmosphere, it will cause atmospheric pollution. Therefore, research on formation mechanism and microphysical characteristics of fog is conducive to deepening people's scientific understanding, thereby improving the accuracy of fog forecasting.

In the autumn of 1959, Okita (1962) used balloons to measure the distribution of fog droplet number concentration and liquid
water content with height in mountain fog in Hokkaido, Japan. In 1970, NASA supported the Cornell Laboratory (CAL) to conduct field observation experiments in New York for the purpose of artificial fog elimination, obtaining the vertical structure of fog microphysical characteristics (Pilié et al.,1975). The earliest fog observation experiment in China took place in Shanghai (Li, 2001). In 1968 and 1969, a fog census was also conducted in the southern provinces of China, and preliminary observations of fog microstructure were made in Yunnan, Guizhou, and Sichuan (Niu et al., 1989). Niu et al. (2010) investigated the
microphysical characteristics of persistent fog, using experimental data from fog observations in Pancheng town, a northern suburb of Nanjing, in winter 2006. Yang et al. (2021) used the microphysical observation data of the Tianjin radiation fog in the winter of 2016/2017 combined with meteorological tower data to reveal the observation facts of the microphysical and size distribution characteristics of fog droplets and discuss fog formation and dissipation mechanisms. Wang et al. (2021) elaborated the physical and chemical characteristics of radiation fog using observation data from December 2019 to February
2020 in the tropical rainforest area of Xishuangbanna.

Given the above, the microphysical characteristics investigation occupies an important position in the study of fog. The relative dispersion is an important physical quantity of the fog microphysical characteristics, and it is a parameter that describes the degree of relative dispersion of the fog droplets size distribution. Besides, the relative dispersion of cloud (fog) droplet
spectrum has been the focus of research in cloud physics for the last two decades (Desai et al., 2019), which can also influence the fog lifetime by affecting the autoconversion threshold function (T) (Lu et al., 2021). It is crucial for the accurate description of cloud droplet microphysics in weather and climate model. Many observational experiments on relative dispersion have been carried out worldwide (Zhao et al., 2006; Berg et al., 2011).Lots of meteorologists have explored the relationship between the relative dispersion and microphysical quantities. For example, Liu and Daum (2002), Rogers and Yau (1989), Yum and
Hudson (2005) used condensation theory to predict that relative dispersion would increase with increasing of cloud droplet number concentration. However, observations have also shown that the relationship between relative dispersion and number concentration is not simply a monotonic increasing or decreasing relationship. It is shown that relative dispersion decreases with increasing number concentration (Lu et al., 2007, 2012; Miles et al., 2000; Pawlowska et al., 2006; Wang et al., 2009). The relationship between relative dispersion and number concentration can also affect the aerosol first effect by changing the

effective radius of cloud droplets (Liu et al., 2002; Lu et al., 2021). Zhao et al. (2006) found that relative dispersion of cloud

droplets gradually converges as the number concentration increases. In addition to the correlation with number concentration,

there is also a correlation between the relative dispersion and the mean-volume diameter of cloud droplet, which determines

the effect of relative dispersion on the Twomey effect (Wang et al. 2023) , while the correlation between the two should also

consider the role of cloud water content (Liu et al., 2008).Martins et al. (2009) analyzed observations from the Amazon region

and found a positive correlation between relative dispersion and mean-volume diameter. However, it has also been shown that

there is a negative correlation between these two as well (Liu and Daum, 2002; Anki et al., 2016).

Therefore, numerous factors influence the relative dispersion evolution, and its evolution pattern has great uncertainty, which

makes the study of cloud and fog microphysics difficult. It has been investigated that droplet collision-coalescence processes

may also have an impact on the spectral dispersion and its evolutionary pattern on the basis of droplet nucleation and

condensation growth. With the T-value is an important parameter for measuring the automatic transformation from cloud to

rain droplet, its numerical magnitude indirectly indicates the strength of the collision-coalescence process in clouds and fog

(Liu et al., 2005,2006). This paper intended to analyze the fog micro-physical characteristics and relative dispersion

characteristics in conjunction with T-value, background aerosol and supersaturation evolutions, so as to deepen the scientific

understanding of the fog micro-physical properties.

## 2 Methods

### 2.1 Introduction of field observation

In the winter of 2019 (November 22, 2019 to February 28, 2020), the fog research team of the NUIST and researchers in the

Key Laboratory of Tropical Forest Ecology conducted a comprehensive field detection experiment in the tropical rainforest

area of Xishuangbanna, China. This work used data sampled by the Fog Monitor in model 120 (FM-120) from the Droplet

Measurement Technologies (DMT) and A 1000XP Wide-Range Particle Spectrometer (WPS-1000XP, MSP Corporation,

USA) during the experiment. Different from FM-100 (Spiegel et al., 2012),the measurement range of the FM-120 is 2-50

μm with a sampling frequency of 1 Hz, hence no data exclusion processing was required for the first bin. The WPS-1000XP

measures the aerosol number-size distribution ($N_a(D)$) with diameter ranging from 10 nm to 10 μm divided into 120 bins and

completes a full spectrum sample every 6 minutes. Due to the lack of visibility observations, this article referred to cloud

criteria, using a number concentration greater than 10 $cm^{-3}$ and a water liquid content greater than or equal to 0.001 g $m^{-3}$ as

the fog criteria, to judge whether it is a fog process from a microphysical perspective (Wang et al., 2021; Lu, et al. 2013).



## 2.2 Calculation

### 2.2.1 microphysical quantities in fog

Based on the fog droplet spectrum ($N_f(D)$) measured by FM-120, the number concentration of fog droplets ($N_f$), mean arithmetic diameter ($MD$), mean-volume diameter ($MVD$), liquid water content ($LWC$), standard deviation of droplet spectrum ($\sigma$), and relative dispersion ($\varepsilon$) were calculated. Where D denotes the droplet diameter of each bin. The calculation formulas are as follows:

$$N_f = \sum_{D_i}^{k} \frac{N_f(D_i)}{PAS \cdot S \cdot \Delta t}, \tag{1}$$

$$MD = \frac{\int N_f(D) \times D \; dD}{N_f}, \tag{2}$$

$$MVD = \left(\frac{\int N_f \times D^3 \; dD}{N_f}\right)^{\frac{1}{3}}, \tag{3}$$

$$LWC = \int \frac{\pi}{6} \rho_w \times N_f(D) \times D^3 \; dD, \tag{4}$$

$$\sigma = \left(\int N_f(D) \times (D - MD)^2 \; dD / N_f\right)^{1/2}, \tag{5}$$

$$\varepsilon = \frac{\sigma}{MD}, \tag{6}$$

Where, $k = 30$, $PAS$ is the airflow velocity in m s$^{-1}$, $S$ is the sampling area in mm$^2$, $t$ is the sampling frequency, and the integration range of the above equation is from 2 to 50 μm. At the same time, in order to eliminate the influence of instrument noise on the acquired data, the FM-120 droplet spectrum information was averaged for 1 min in this paper.

Since aerosols complete a full spectrum sampling every 6 min, each aerosol particle data was considered to represent the
average state within 6 min for the calculation. In order to combine the aerosol data with the fog data for analysis, the fog data within one aerosol sampling time were therefore averaged, and the individual microphysical quantities in the fog were recalculated.

### 2.2.2 Visibility

For facilitating the study of fog classification, the visibility data in fog observed in Xishuangbanna in this paper were calculated
using the $N_f(D)$ information. The visibility and extinction coefficients were calculated as follows.

$$V = \frac{-\ln \alpha}{\beta}, \tag{7}$$

In the above equation,





$$\beta = \pi \sum_{i=1}^{k} Q_{ext} n_i(r_i) r_i^2 , \tag{8}$$

Among them,

$$Q_{ext} = 2 - \frac{4}{\rho} \sin \rho + \left(\frac{2}{\rho}\right)^2 (1 - \cos \rho), \tag{9}$$

In the above equation, α is the contrast threshold, usually equal to 0.02. $\beta$ is the extinction coefficient in $km^{-1}$. $Q_{ext}$ is the meter scattering coefficient. $r_i$ is the radius of the i-th fog droplet. $n_i(r_i)$ is the fog droplet number concentration at the radius. Where $\rho = 2x(m-1)$, for water droplets m=0.75. And $r$ is the radius, which is usually considered to be 0.5.

### 2.2.3 Supersaturation

Based on the calculation method of Petters and Kreidenweis (2007), Shen et al. (2018) and Mazoyer et al. (2019), since droplets are the product of aerosol activation under specific supersaturation conditions, the supersaturation ($SS$) can be calculated indirectly with known information about aerosols and droplets.

$$D_c = \left(\frac{3kD_a^3}{A}\right)^{\frac{1}{2}} , \tag{10}$$

Where,

$$A = \frac{\frac{4\sigma_s M_w}{a}}{RT_{em}\rho_w} , \tag{11}$$

In the above equation, $\sigma_{s/a}$ is the droplet surface tension (assumed to be pure water, 0.0728), $M_w$ is the molar mass of water, $R$ is the gas constant, $T_{em}$ is the ambient atmospheric temperature measured in real time using FM-120, $D_c$ is the diameter of the activated droplet, and $D_a$ is assumed to be the critical activation diameter of the dry aerosol. Referring to measurements of aerosol hygroscopicity in the Amazon rainforest region (Pöschl et al., 2010; Wang et al., 2021), $k$ was assumed to be 0.15. where A can be simplified (Seinfeld and Pandis, 2016) as,

$$A \cong \frac{0.66}{T_{em}} , \tag{12}$$

A as a function of temperature $T_{em}$ in μm. $D_a$ was calculated by referring to Wang et al.(2021). The calculation of $D_a$ was done by making a cycle from large to small with the central diameter $D_a$ of each of the 120-bin aerosol spectra, finding the two $D_a$ corresponding to when the integration of the droplet spectrum concentration in the previous cycle was larger than the integration of the aerosol spectrum concentration, and when the opposite sign was in the next cycle, and obtaining the integration when it is equal by interpolation, and after calculating, the corresponding of $D_c$, so that the supersaturation $SS_c$ can be calculated, thereby obtaining $D_a$ and its corresponding $D_c$. Finally, according to the reciprocal relationship between the critical $SS(SSc)$ and $D_c$.



$$SS_c = \frac{200A}{3D_c},$$ (13)

### 2.2.4 Autoconversion threshold function

In order to study the collision-coalesces process in fog, an autoconversion threshold function ($T$) was introduced. $T$ is an important microphysical quantity describing the intensity of the collision-coalesces between droplets within clouds and fog. The calculation derived by Liu et al. (2005) as follows:

$$T = \frac{P}{P_0} = \left[\frac{\int_{r_c}^{\infty} r^6 n(r)\, dr}{\int_0^{\infty} r^6 n(r)\, dr}\right]\left[\frac{\int_{r_c}^{\infty} r^3 n(r)\, dr}{\int_0^{\infty} r^3 n(r)\, dr}\right],$$ (14)

where $r$ is the radius of the droplet, b is the number of cloud droplets per unit volume per unit radius, and $r$ is the critical radius of the $T$. the analytical expression derived by Liu et al. (2004) is as follows:

$$r_c \approx 4.09 \times 10^{-4} \beta_{con}^{\frac{1}{6}} \frac{N_f^{\frac{1}{6}}}{LWC^{\frac{1}{3}}},$$ (15)

$\beta_{con} = 1.15 \times 10^{23}\ \mathrm{s}^{-1}$, is an empirical constant. $T$ varies from 0 to 1, with larger values of $T$ indicating higher collection efficiency.

### 3 Results and analysis

Based on the start and end times of the 19 fog events measured during the XSBN-FOG-2019 experiment. These 19 fog events have typical radiation fog characteristics (Wang et al., 2021). Referring to the national standard "Fog Forecasting Level" (General Administration of Quality Supervision, Inspection and Quarantine of the People's Republic of China and China National Standardization Management Committee, 2012), fog was classified according to visibility. Table 1 shows the average values of different microphysical characteristics for fog with different visibility. The average value of $T$ for the 19 fog samples was 0.38, $N_f$ was 48.55 cm$^{-3}$, LWC was 0.078 g m$^{-3}$, ε was 0.56. As the fog intensity increased, all physical quantities increased. As shown in Fig. 1, $T$, $N_f$, $MVD$, and $LWC$ all increased as $V$ decreased.

Figure 2 shows the relationship between the fog microphysical quantities and the fog time percentage for the 19 fog processes. The proportion of fog time indicates the fog time as a percentage of the overall process duration, and a larger proportion indicates a more continuous fog. $N_f$, $LWC$, $MVD$, and ε all increase with increasing fog time percentage. This is due to the fact that when the fog is more continuous, the fog persists longer and is relatively deeper.





**Table 1: Mean values of fog microphysical characteristic quantities at different visibilities**

| Fog Types | $N_f(\text{cm}^{-3})$ | $MVD(\mu\text{m})$ | $LWC(\text{g m}^{-3})$ | $T$ | $\varepsilon$ |
|---|---|---|---|---|---|
| Heavy fog (500m-1000m) | 44.15 | 14.40 | 0.052 | 0.34 | 0.64 |
| Dense fog (200m-500m) | 59.35 | 17.09 | 0.118 | 0.62 | 0.68 |
| Strong dense fog(<200m) | 66.65 | 20.23 | 0.256 | 0.86 | 0.72 |

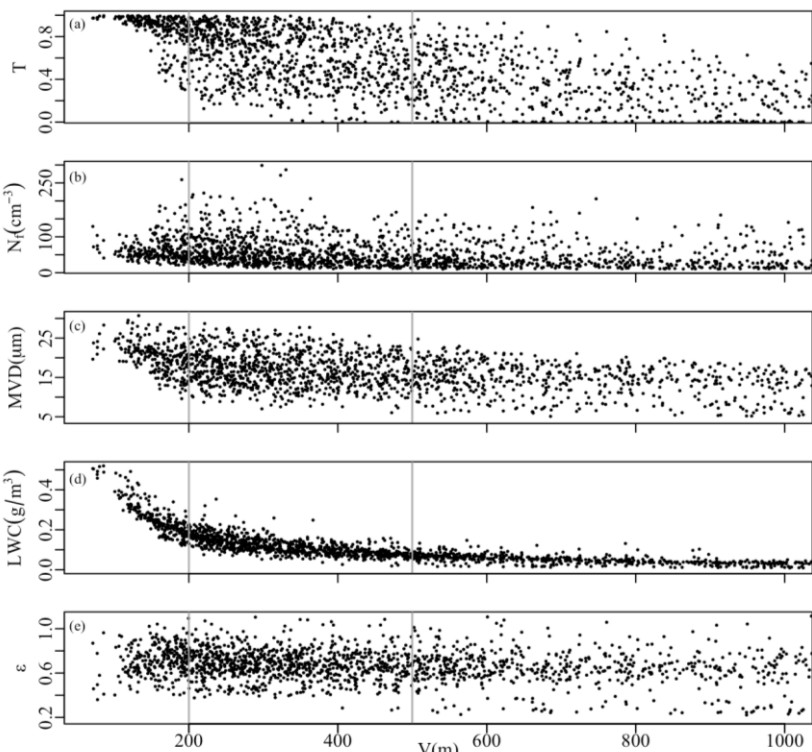

**Figure 1: Variations of autoconversion threshold ($T$), number concentration ($N_f$) and volume mean diameter ($MVD$) of fog droplets, liquid water content ($LWC$), and relative dispersion ($\varepsilon$) of fog droplet spectrums with visibility ($V$)**





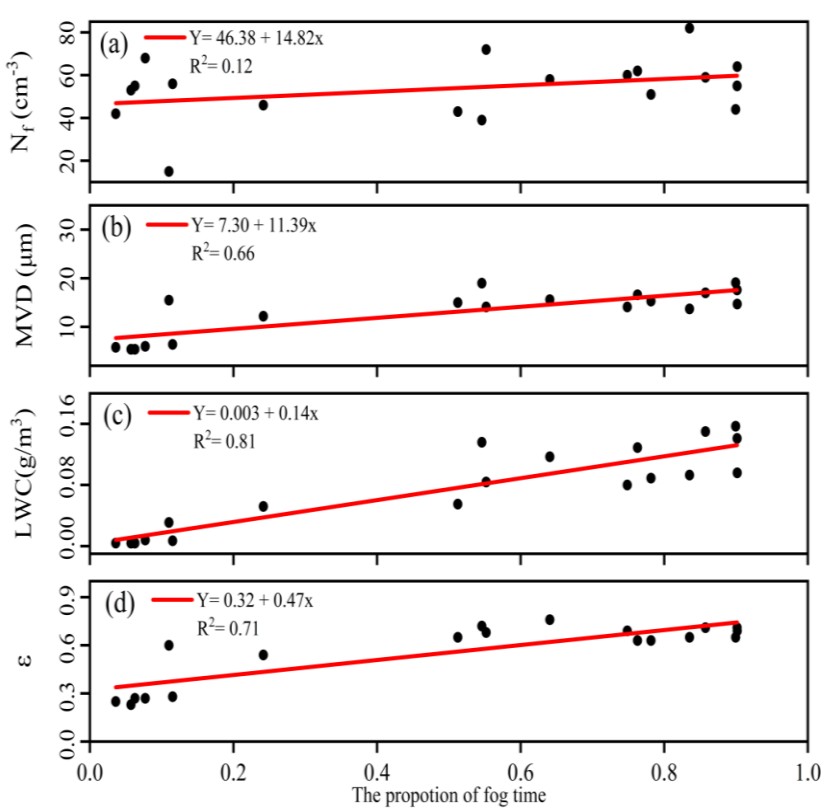

**Figure 2 Variations of the mean values of number concentration ($N_f$) and volume mean diameter ($MVD$) of fog droplets, water content ($LWC$), and relative dispersion (ε) of fog droplet spectrums, with respect to the proportion of fog time ,with red solid lines are linear fitted lines**

### 3.1 The relationship between relative dispersion and fog microphysical characteristics

It is shown in Fig. 3a that the relationship between ε and $MVD$ in fog is not a simple monotonic relationship. For $MVD$ less than 15 μm, ε and $MVD$ are positively correlated, and for $MVD$ greater than 15 μm, the relationship becomes negative. Lu et al. (2020) linked the relationship between these two quantities to microphysical processes, pointing out that when $MVD$ and ε are positively correlated, droplet condensation and activation processes occur simultaneously, and the changes in ε and $MVD$ are consistent. When $MVD$ is negatively correlated with ε, only condensation process or weak activation exists in the cloud. Therefore, if the fog droplet grows mainly through the condensation process, there is a limitation in its size growth, while the small droplets grow up by condensation process, and the weak activation of droplets leads to the small droplets cannot be replenished in time, thus the $N_f(D)$ becomes narrower and the ε decreases.

The microphysical characteristics of fog are similar to those of stratiform clouds (Wang et al., 2023), and research findings on stratiform clouds are applicable to fog droplets. In Xishuangbanna, there are differences in the relationship between ε and



$MVD$ under different collision and coalescence intensities (Fig.3a). When $T < 0.4$, $\varepsilon$ increases and then decreases with $MVD$. When $0.4 < T < 0.8$, the decreasing trend of $\varepsilon$ with $MVD$ is significant. As $T$ increases to above 0.8, the decreasing trend of $\varepsilon$ with $MVD$ weakens. During the fog process, not only the condensation and activation of fog droplets affect $\varepsilon$ , but also the collision and coalescence processes between fog droplets have an impact on $\varepsilon$ of fog spectrum, which can dominate the negative correlation between $\varepsilon$ and $MVD$, should not be ignored.

As shown in Fig. 3b, in the context of low $T$ values ($T \le 0.4$), if both condensation and activation processes occur simultaneously, $\varepsilon$ also increases with the increase of $LWC$. As shown in Fig. 4, when $MVD$ is less than 15 μm, $\varepsilon$ increases with the increase of $LWC$, with both fog droplet activation and condensation are active in this process. However, after $MVD$ exceeds 15 μm, $\varepsilon$ does not continue to increase with the increase of $LWC$, indicating a lower level of activity in the fog droplet nucleation process. This further suggests that the activation process of fog droplet weakens when MVD of fog droplet exceeds 15 μm.

Figure 5 shows the scatter plot of $\varepsilon$ and $N_f$ for different ranges of $T$ values. From Fig. 5a and Fig. 5b, it can be seen that when $T$ is less than or equal to 0.4, $\varepsilon$ converges as $N_f$ increases. However, for $T$ values greater than 0.4, there is a tendency for $\varepsilon$ to increase with an increase in $N_f$. There are two possible scenarios for a lower $N_f$, either the fog droplet activation is limited, or the coalescence process is more active, leading to a reduction in $N_f$. Therefore, in situations where the coalescence process is more active, $\varepsilon$ decreases with a decrease in $N_f$.

In order to analyze in depth, the probability density of droplet size and concentration in different $T$ value ranges were analyzed (Fig. 6). When $T \le 0.2$, the droplets are mainly distributed in the size of less than $12\mu m$ (Fig. 6a). When $0.2 < T < 1.0$ (Fig. 6b-e), the $N_f$ of larger droplets increases and smaller droplets gradually decreases. As shown in Fig. 6e, for $0.8 < T \le 1.0$, the $N_f$ of droplets with sizes larger than 12 μm increases to the maximum, and the $N_f(D)$ widens towards the larger droplet end. The main reason may be that under larger T values, the coalescence process of droplets is active, leading to excessive consumption of smaller droplets and generation of larger droplets. It can be seen that when $T \le 0.4$, there are generally not many large droplets, and the growth of $N_f$ mainly relies on the nucleation process to increase the $N_f$ of smaller droplets, resulting in slow condensation growth and a relatively narrow $N_f(D)$. When $T > 0.4$, the coalescence between droplets increases the occurrence probability of large droplets, and the consumption rate of small droplets accelerates. Therefore, increasing the $N_f$ is beneficial to broaden the $N_f(D)$ in this situation, and there is a positive correlation between $\varepsilon$ and $N_f$.



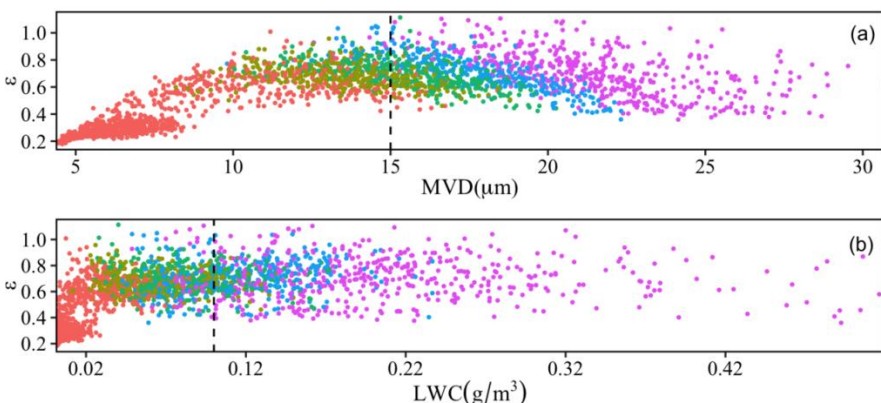

**Figure 3: Variation of relative dispersion ($\varepsilon$) with volume mean diameter ($MVD$) and liquid water content ($LWC$),**

**(a) black dashed line for MVD=15 µm; (b) black dashed line for LWC=0.1 g m$^{-3}$**


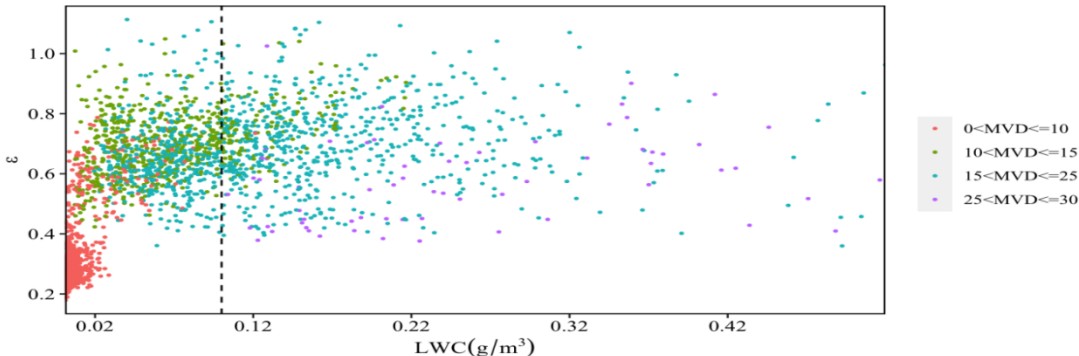

**Figure 4: Relationship between water content ($LWC$) and relative dispersion ($\varepsilon$) (black dashed line LWC=0.1 g m$^{-3}$)**





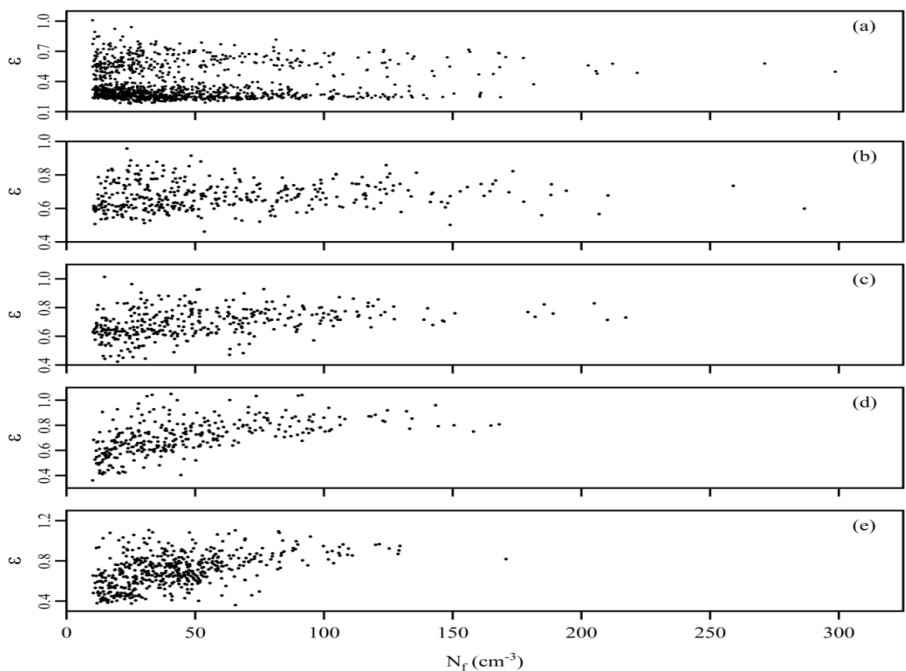


**Figure 5: Scatter plot of relative dispersion ($\varepsilon$) versus number concentration ($N_f$) for (a) T≤0.2;(b) 0.2<T≤0.4;(c) 0.4<T≤0.6;(d) 0.6<T≤0.8;(e) 0.8<T≤1.0**

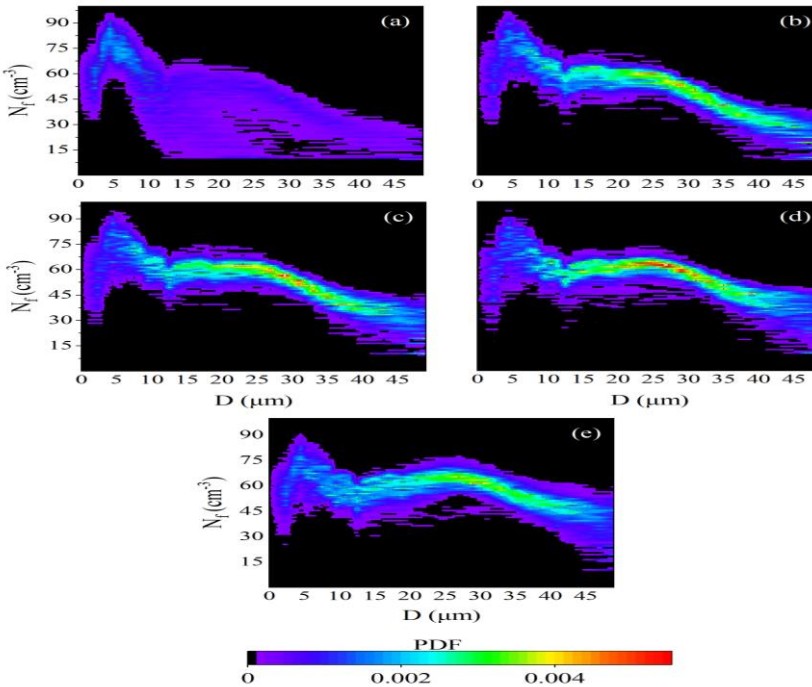



**Figure 6: Fog droplet particle size, concentration probability density function (a) $T \leq 0.2$; (b) $0.2 < T \leq 0.4$; (c) $0.4 < T \leq 0.6$;**
**(d) $0.6 \leq T < 0.8$; (e) $0.8 < T \leq 1.0$**

### 3.2 Relationship between T value and fog microphysical quantities

As mentioned earlier, $T$ reflects the activity of the collision-coalescence of fog droplets. In the low T-value background, there is a limitation of particle size increase due to droplet condensation, resulting in narrower droplet spectrums. As $T$ increases, the collision-coalescence process of fog droplets becomes more and more active, so the $\varepsilon$ may be affected by the nucleation,
condensation, and coalescence processes of fog droplets. Therefore, it is necessary to consider the evolutions of $T$ in fog process.

Figure 7a and b show that $T$ has an increasing and then constant relationship with $MVD$ and $LWC$. The relationship between $T$ and $LWC$ indicates that the strength of the coalescence process is determined by the condition of the fog droplet condensation
process. Condensation growth leads to a broadening of the fog droplet spectrum and the occurrence of the coalescence process. A larger $MVD$ makes the coalescence process more likely to occur. However, there are still differences in the increasing trend of $T$ with $LWC$ and $MVD$. When $MVD$ is small, the rate of increase in $T$ is slower, indicating that during this stage, the condensation growth gradually allows the coalescence process to occur and become active. In the later stage, with the occurrence of the coalescence process, $MVD$ increases rapidly, further enhancing the coalescence process.


Compared to $MVD$ and $LWC$, the relationship between $T$ and $N_f$ or $\varepsilon$ is relatively complex. When $N_f$ is less than or equal to 100 cm$^{-3}$, there is no significant relationship between $T$ and $N_f$. However, when $N_f$ is greater than 100 cm$^{-3}$, $T$ tends to decrease with an increase in $N_f$. The enhancement of the coalescence process has a negative effect on $N_f$, which is related to the interaction between the coalescence, and activation-condensation of fog droplets. If the former is stronger than the latter,
the $N_f$ decreases, and vice versa, the $N_f$ increases. However, if their effects cancel each other out, in the absence of other external forces, the $N_f$ remains unchanged. Thus, low values of $N_f$ may correspond to different situations, which may be one of the main reasons why $T$ values are not significantly related to them when $N_f$ is less than 100 cm-3.

According to Fig 7d, When $\varepsilon \leqslant 0.4$, $T$ is generally less than 0.1. When $\varepsilon > 0.4$, $T$ initially increases with $\varepsilon$ and then begins to
spread towards larger and smaller values. The larger the $T$, the more obvious the spreading phenomenon. To further analyze the relationship between $T$ and $\varepsilon$, the concentration and size probability density distribution of fog droplets with different $T$ and $\varepsilon$ are analyzed (Fig. 8).

Combining the relationship between $\varepsilon$ and $T$ in Fig 7d, as shown in Fig 8, the relationship of $\varepsilon - T$ in $T > 0.4, 0.4 < \varepsilon \leq 0.7$
(Fig. 8c) is negative, and the rest are positive. For $T > 0.4, 0.4 < \varepsilon \leq 0.7$, the $N_f$ of $2 \sim 12 \mu m$ significantly smaller than the other cases. The reason for this may be that the collision-coalescence process is active and the nucleation process cannot





replenish the small droplets consumed by the collision-coalescence process in time, and the $N_f(D)$ will broaden toward the large droplet end, so the stronger the collision-coalescence process is, the smaller the $\varepsilon$ is.

As can be seen from Fig. 8c and f, even with the same range of $T$ values, there is a significant variation in the $\varepsilon$ in which may be related to the aerosol concentration in the environment where the instrument was located (Chandrakar et al., 2018). It is evident that a larger $\varepsilon$ corresponds to a wider range and higher probability of particle number concentrations greater than 12 µm. At the same time, a lower $N_f$ in the range of 2-12 µm may likely result in lower $\varepsilon$ of the fog (Fig. 8d).

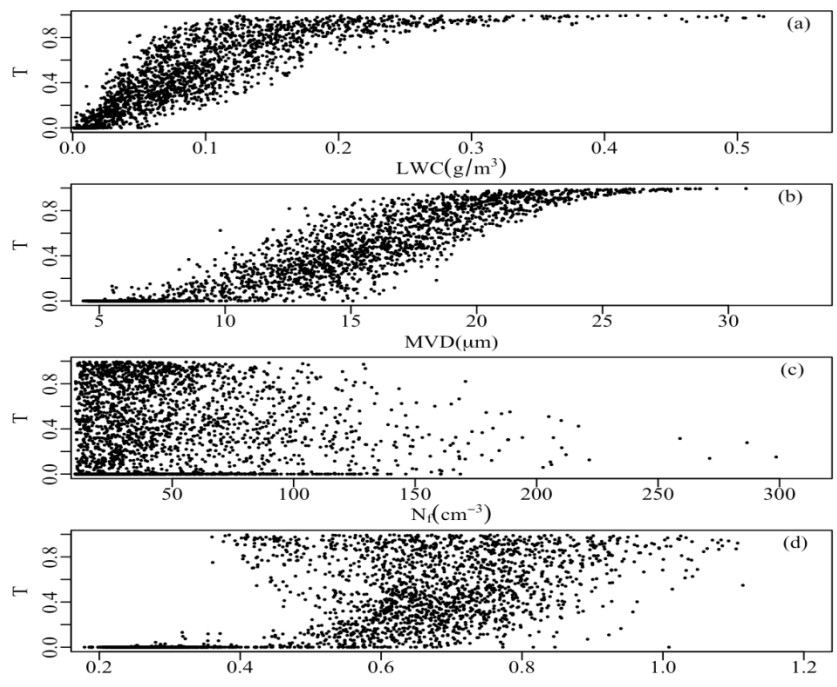

**Figure 7: Scatter plot of autoconversion threshold ($T$) and water content ($LWC$), volume mean diameter ($MVD$), number concentration ($N_f$), relative dispersion ($\varepsilon$)**



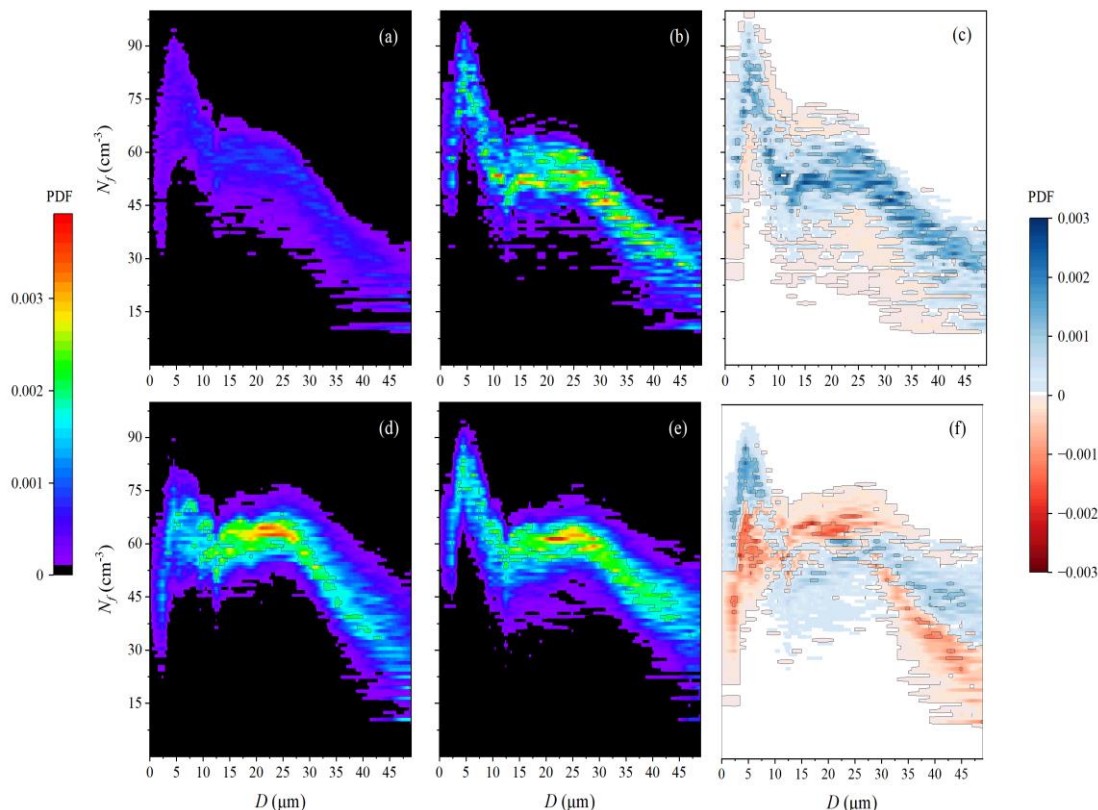

**Figure 8: Probability density function of fog droplet size and concentration for (a)$T \leq 0.4, 0.4 < \varepsilon \leq 0.7$;(b)$T \leq 0.4$,$\varepsilon > 0.7$;(c)difference of $T \leq 0.4$,$\varepsilon > 0.7$ and $T \leq 0.4, 0.4 < \varepsilon \leq 0.7$;(d)$T > 0.4, 0.4 < \varepsilon \leq 0.7$;(e)$T > 0.4, \varepsilon > 0.7$;(f)difference of $T > 0.4, \varepsilon > 0.7$ and $T > 0.4, 0.4 < \varepsilon \leq 0.7$**


In summary, the correlation between $\varepsilon$ and $T$ is related to the $N_f$ in the range of 2 μm to 12 μm. The coalescence process can reduce the number of small droplets and broaden the droplet spectrum towards the larger droplet end. If the smaller droplets are not replenished, $\varepsilon$ decreases with increasing $T$, whereas $\varepsilon$ and $T$ are positively correlated when the smaller droplets are replenished. Although $\varepsilon$ is related to $T$, high $\varepsilon$ values can still exist in the background of low $T$ values, and similarly, low $\varepsilon$
values can exist in the background of high $T$ values, both of which are related to the activation of fog droplet at the location.

In addition, the occurrence probability and $N_f(D)$ of fog droplets with diameters from $20\mu m$ to $28\mu m$ reflect the magnitude of $T$ from a lateral perspective. A higher $N_f$ value for fog droplets with diameters ranging from 20 μm to 28 μm indicates a stronger $T$ value.




### 3.3 Relationship between supersaturation and fog microphysical characteristics

Under atmospheric supersaturation conditions, aerosol particles can be activated and form fog droplets, thereby indirectly affecting the $\varepsilon$. In this study, the $SS$ obtained from the dry aerosol spectrum will be used to analyze its impact on fog

characteristics. The fog droplet concentration increases with increasing $SS$ (Fig. 9a).

As shown in Fig. 9b, when the $MVD$ is less than 10 μm, there is no significant correlation between $SS$ and $MVD$. However, when $MVD$ is greater than 10 μm, $MVD$ increases initially and then decreases with increasing $SS$. Based on Fig. 9c and Fig. 9e, it can be seen that $LWC$ and $T$ also exhibit similar relationships with $SS$. The average value of $D_a$, calculated by taking

into account the $\varepsilon$ less than or equal to 0.4, is approximately 0.3 μm. For $\varepsilon$ greater than 0.4, the average value of $D_a$ is approximately 0.25 μm. Therefore, the larger the aerosol activation particle size, the narrower the fog droplet spectrum, resulting in a small $\varepsilon$ that does not vary much with $SS$. As shown in Fig. 9d, there is no significant relationship between $SS$ and $\varepsilon$ when $\varepsilon \leq 0.4$. However, when $\varepsilon > 0.4$, $\varepsilon$ initially increases and then decreases with $SS$.

From Fig. 10, it can be seen that as $SS$ increases, the $N_f(D)$ of fog droplets smaller than $10 \mu m$ increases. As $SS$ increases, the $\varepsilon$ increases due to the combined effects of cloud droplet condensation growth and activation. With further increases in $SS$, the activation of droplets intensifies, leading to a decrease in the $MVD$ of the droplets. What's more, when $SS$ exceeds 0.12 %, the $N_f$ of fog droplets in the range of 30 μm to 50 μm decreases, may also due to gravity settling, leading to a decrease in $MVD$ and $LWC$, and subsequently a decrease in $T$.

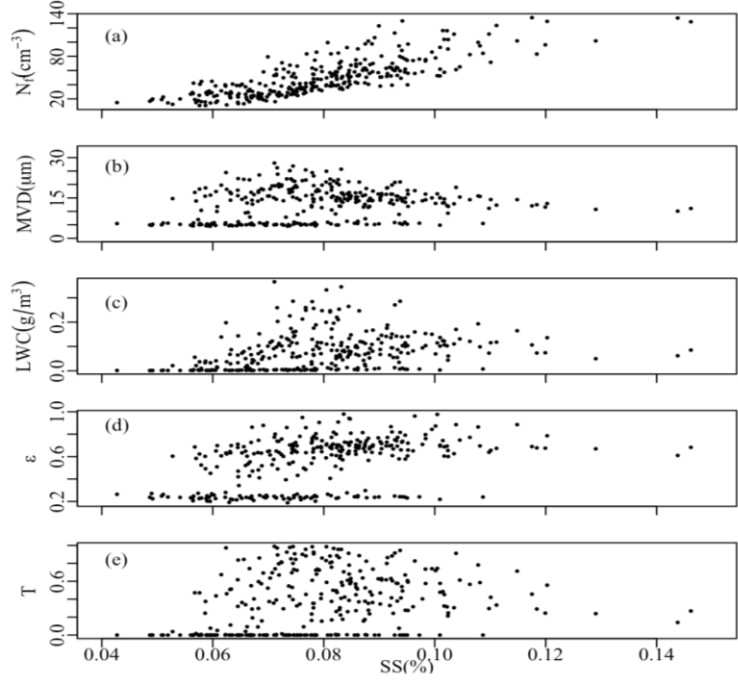




**Figure 9: Scatter plots of supersaturation ($SS$), number concentration ($N_f$) and volume mean diameter ($MVD$) of fog droplet, water content ($LWC$), dispersion ($\varepsilon$), and auto-conversion threshold ($T$) in fog.**

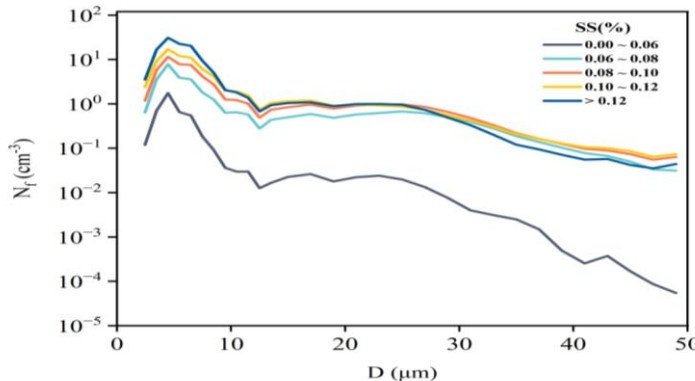

**Figure 10: Average spectrum of fog droplets at different supersaturation levels**

Probability distribution of fog droplet size and concentration for both $\varepsilon \leq 0.4$ (Fig. 11a) and ε>0.4 (Fig. 11c) reveals that in the case of ε>0.4, the presence of large droplets (>12 μm) in fog significantly increases, and the $N_f(D)$ extends towards the larger droplet end (>28 μm).It is shown that, fog droplets are mostly in the small droplet range and primarily grow by condensation with $\varepsilon \leq 0.4$(Fig. 11a). Comparing difference of $\varepsilon > 0.4$ and $\varepsilon \leq 0.4$ (Fig. 11e), it can be seen that in $\varepsilon > 0.4$ (Fig. 11e), the aerosol number concentration and probability is less in the all range compared to $\varepsilon \leq 0.4$ (Fig. 11d). In this

case, due to the high aerosol concentration, the concentration of activated fog droplets is also high, resulting in a slow growth rate of the droplets and a smaller relative dispersion. This is the reason that, in Fig. 9d, when $\varepsilon \leq 0.4$, the $\varepsilon$ remains essentially constant as the $SS$ increases. This leads to fog droplets primarily distributed in the range of $2\mu m$ to $12\mu m$, and their growth depends on the condensation process, with limited growth rates. Increasing $SS$ only increases the $N_f$ of fog droplets. When ε>0.4, the $N_f(D)$ of fog droplets larger than 12 μm significantly increases. As $T$ increases, although the collision-coalescence

process consumes small droplets, the increase in $SS$ still leads to an increase in the number of small droplets. If the number of small fog droplets consumed by the collision-coalescence and condensation process is replenished by the fog droplets generated by the activation process, leading to an increase in ε. However, when the $SS$ increases to 0.1 %, $\varepsilon$ starts to decrease. This is because if there are too many small fog droplets, it will result in a decrease in the $MVD$. What's more, there maybe sedimentation of large droplets with $SS$ larger than 0.12 %, leading to a decrease in the $\varepsilon$ of the fog droplet spectrum. It is

shown that $SS$ and aerosols fundamentally affect the magnitude of the $T$ and $\varepsilon$.





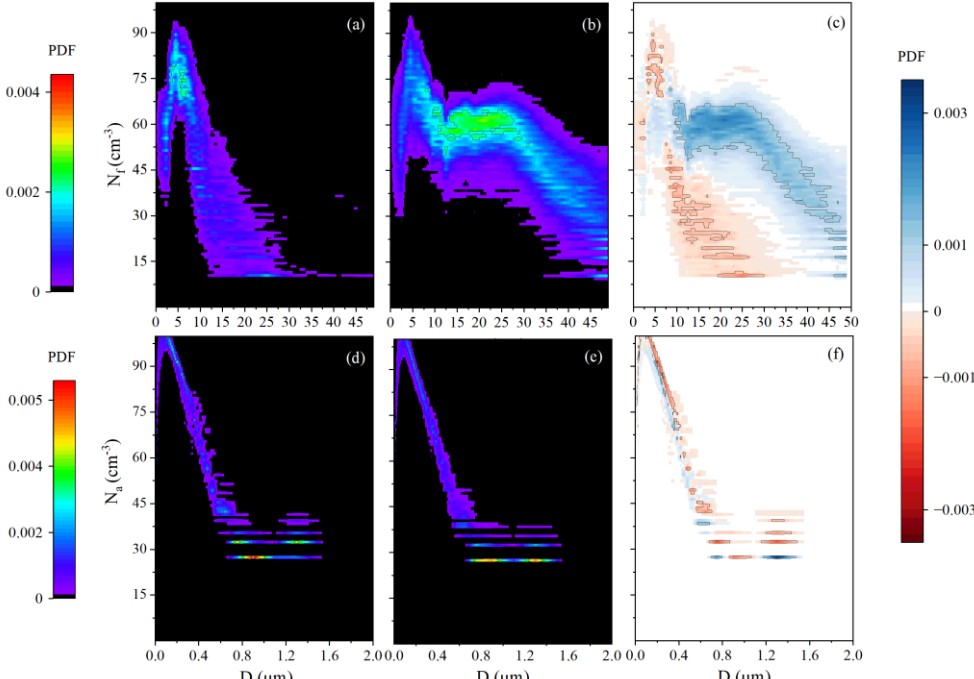

**Figure 11 Size and concentration probability density (PDF) diagrams of fog droplet size distribution for (a) $\varepsilon \leq 0.4$; (b) $\varepsilon > 0.4$; (c) difference of $\varepsilon > 0.4$ and $\varepsilon \leq 0.4$ and of aerosol particles size distribution for (d) $\varepsilon \leq 0.4$; (e) $\varepsilon > 0.4$; (f) difference of $\varepsilon > 0.4$ and $\varepsilon \leq 0.4$**

## 4 Discussion

This study explores the evolution characteristics of fog microphysics based on 19 fog observation data and found that the relationship between $\varepsilon$ and $MVD$ is similar to the findings of Lu et al. (2020), indicating the existence of a critical value for the $MVD$. Above and below the critical value, the relationship between $\varepsilon$ and $MVD$ undergoes changes. It is worth mentioning that in the study by Rui et al. (2022), an increase in the $T$ weakens the negative correlation between $\varepsilon$ and $MVD$, and strengthens the positive correlation. However, in this study, it was found that an increase in the $T$ enhances the negative correlation between $\varepsilon$ and $MVD$. The main reason for the increasing and then converging trend of fog droplet spectrum $\varepsilon$ and $LWC$ is that the $LWC$ is easily influenced by fog droplet activation and condensation growth, weakening the negative correlation between $\varepsilon$ and $LWC$. Previous research results have shown that Liu and Daum (2002), Rogers and Yau (1989), Yum and Hudson (2005) used condensation theory to predict a positive correlation between $\varepsilon$ and $N_f$. However, there are also observations indicating a negative correlation between $\varepsilon$ and $N_f$ (Yum et al., 2005; Pawlowska et al., 2006; Lu et al., 2007), and the change in $\varepsilon$ with increasing $N_f$ is positive or shows a gradually converging trend (Zhao et al.2006). In this study, both of the above phenomena were observed simultaneously. When T is small and the $N_f(D)$ is narrow, the $\varepsilon$ gradually converges with an increasing $N_f$. When $T$ becomes larger and the $N_f(D)$ widens, the $\varepsilon$ shows a positive correlation with the $N_f$.

Therefore, the relationship between $\varepsilon$ and $N_f$ needs to be discussed separately based on different situations. The relationship between $\varepsilon$ and $T$ as well as $SS$ is more complex. Additionally, the correlation between $\varepsilon$ and $T$ and $SS$ depends to some extent on the aerosol size distribution and the number of activated aerosols.

## 5 Conclusions

In this study, we used the droplet and aerosol particle spectrum data obtained during 19 radiation fog observation experiments
conducted in Xishuangbanna in winter 2019 to investigate the evolution patterns and mechanisms of radiation fog microphysical properties in the region and obtain the following conclusions:

(1) There are significant differences in the correlation between various microphysical quantities and the $\varepsilon$ under different $T$ values. At $T$ below 0.4, $LWC$ and $MVD$ are positively correlated with $\varepsilon$, mainly due to the dominate condensation and
activation of fog droplet. Fog droplet growth primarily depends on the condensation process, leading to $\varepsilon$ converging with increasing $N_f$ in this situation. When $T$ is greater than 0.4, there is no significant correlation between $LWC$ and $\varepsilon$, and the effects of fog droplet condensation, coalescence, and activation processes all need to be considered. In this condition, the collision-coalescence and condensation processes are stronger than the activation process, thus $MVD$ is negatively correlated with $\varepsilon$. The active coalescence process tends to widen the $N_f(D)$ towards the larger droplet end, increase $N_f$, and facilitate an
increase in $\varepsilon$, so $\varepsilon$ is positively correlated with $N_f$. Therefore, the strength of the coalescence process has a certain influence on the change rule of the $\varepsilon$.

(2) Increasing the $T$ can increase the $MVD$ and $LWC$ in the fog, and the impact on the $N_f$ is limited by the activation process. The correlation between $\varepsilon$ and T is constrained by the $N_f$ in the range of 2 μm to 12 μm of the fog, and they do not have a
simple linear relationship. In addition, the quantity of large droplets indirectly reflects the strength of coalescence. More large droplets indicate stronger coalescence, while fewer large droplets indicate weaker coalescence. If both the coalescence process and the activation process are active, it will increase the $\varepsilon$ in the fog. Otherwise, a strong coalescence process will result in low $\varepsilon$.

(3) As the $SS$ increases, the $N_f$ of small droplets in the fog also increases, thereby changing the variation of microphysical quantities that affect the $\varepsilon$. In addition, if the initial nucleated $N_f(D)$ is narrow, and the $SS - \varepsilon$ relationship is not obvious, with little change in $\varepsilon$ with $SS$. After the $N_f(D)$ widens, the $SS - \varepsilon$ relationship strengthens, and the $\varepsilon$ increases and then decreases with increasing $SS$. When the $SS$ is greater than 0.12 %, the $N_f$ of droplets in the range of 30 μm to 50 μm decreases, and larger droplets may settle due to gravity, resulting in a decrease in $\varepsilon$ with increasing $SS$.


This study is mainly based on observations of fog microphysics, and the reasons for the reduction of larger droplets at high *SS* have not been conclusively determined. Future research should combine numerical simulations to further investigate the underlying mechanisms. Additionally, the findings of this study may also be applicable to inland radiative fog processes in relatively clean areas such as Xishuangbanna in China.

**Data availability**

The data are available from the authors upon request.

**Author contributions**

Xiaoli Liu and Zhenya An shaped the concept of this study and refined the approach during extensive discussions. Zhenya An and Xiaoli Liu carried out the data analysis. Zhenya An prepared the figures and wrote the initial draft, which was subsequently 390 refined by both authors.

**Conflict of Interest**

The authors declare that the research was conducted in the absence of any commercial or financial relationships that could be construed as a potential conflict of interest.

**Acknowledgments**

In addition, we acknowledge the High Performance Computing Center of Nanjing University of Information Science and Technology for their support of this work.

**Funding**

This work was supported by the National Natural Science Foundation of China (Grant Nos. 42061134009 and 41975176).

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
