# Peer review of "Observational study of factors influencing the dispersion of warm fog droplet spectrum in Xishuangbanna, China"

_EGUsphere, 2023_

## Referee Comment (RC1)

First review of the manuscript.

Observational study of factors influencing the dispersion of warm fog droplet spectrum in Xishuangbanna, China

by Zhenya An and Xiaoli Liu

Marie Mazoyer, Météo-France

The paper titled "Observational study of factors influencing the dispersion of warm fog droplet spectrum in Xishuangbanna, China" by An et al., aims to study the characteristics of fog microphysical properties by considering 19 fog cases observed in a tropical rainforest. Their study mainly focuses on the relationship between self-conversion and the relative dispersion of fog droplet spectra. Additional emphasis is placed on the relationship of supersaturation with the concentration of fog droplets. I think this study is of great interest given the current challenges of forecasting and modeling fog and associated microphysics. It gives a relevant description of microphysical processes occurring during fog evolution.

However, I feel that clarification of the paper's objectives needs to be made. In this way, the novelty of this work must be emphasized. The presentation of the measurement campaign and the cases studied is completely necessary before considering publication of this work. The introduction should also focus on fog microphysical processes, their impact on the fog life cycle and their consideration in the numerical model. A summary effort is really necessary on your results section for better reading. The concentration of aerosols is sometimes commented on but no plot presents it. The captions need to be rewritten much more explicitly for the description of the figures. Finally, a conceptual diagram ("handmade" graphic) could be very interesting to conclude the article.

Given these points, I'm really hesitant between rejection and major revision. As your article could be significantly improved with additional work, I would suggest a major revision.

Introduction:

-A review of fog processes affecting fog during its evolution could be nice.

-Explain why we need a better description of fog microphysical processes.

-A few words about numerical modeling might be interesting.

-I really appreciate the presentation of the measurement campaign but it lacks international references. On the campaigns and results C-FOG (Gultepe, 2021), LANFEX (Price, 2019), SOFOG-3D (Burnet, 2020) or WIFEX (Ghude, 2023), among others.

-A deeper focus on the relevance of T and $\varepsilon$ for fog description could provide a better understanding of the article and promote their use for future studies in the community. A "handmade" spectra plot could be useful for this task and help the reader understand how variations in T and $\varepsilon$ are related to fog microphysical processes.

-Then, new questions in relation to previous studies Zhao, 2013 for example and among others must be pointed out.

Methods:

- Present the campaign, the instruments and the cases studied. Or make references to any article that has already featured it.

- Supersaturation section: Petters and Kredenweis (2007) (the reference is missing in your reference session) indicates that the formula you used for A cannot be used for kappa < 0.2. As you use it for a kappa=0.15, a discussion is in order. A diagram would be welcome for a better understanding of the method used. See Mazoyer (2009) for example.

Results and analysis:

-Present the figures and the objectives of the figures before commenting on them.

-The captions are not complete enough.

-A comparison of your results with previous studies is sometimes missing.

-Figure 3, you comment on T, but where is T? The color legend is missing.

-Figure 8 and 11, your comments on the concentration of aerosols are very interesting but must be documented with an aerosol concentration plot for example.

Conclusions:

-Please re-introduce the objectives

-Please comment your findings on the processes rather than re-presenting your findings on direct MVD, LWC, T, epsilon,… relationships.

-Please draw the most important conclusions and implications for fog forecasting and modeling.

-A conceptual diagram ('handmade' graphic) could be very nice to conclude the article

References :

-Burnet, F., Lac, C., Martinet, P., Fourrié, N., Haeffelin, M., Delanoë, J., ... & Vié, B. (2020, May). The SOuth west FOGs 3D experiment for processes study (SOFOG3D) project. In EGU General Assembly Conference Abstracts (p. 17836).

-Ghude, S. D., Jenamani, R. K., Kulkarni, R., Wagh, S., Dhangar, N. G., Parde, A. N., ... & Rajeevan, M. (2023). WiFEX: Walk into the Warm Fog over Indo-Gangetic Plain Region. Bulletin of the American Meteorological Society, 104(5), E980-E1005.

- *Gultepe, I., Heymsfield, A. J., Fernando, H. J. S., Pardyjak, E., Dorman, C. E., Wang, Q., ... & Wang, S. (2021). A review of coastal fog microphysics during C-FOG. Boundary-Layer Meteorology, 181, 227-265.*

-*Mazoyer, M., Burnet, F., Denjean, C., Roberts, G. C., Haeffelin, M., Dupont, J. C., & Elias, T. (2019). Experimental study of the aerosol impact on fog microphysics. Atmospheric Chemistry and Physics, 19(7), 4323-4344.*

-*Petters, M. D. and Kreidenweis, S. M.: A single parameter representation of hygroscopic growth and cloud condensation nucleusactivity, Atmos. Chem. Phys., 7, 1961–1971, doi:10.5194/acp-7-1961-2007, 2007.*

-*Price, J., Lane, S., Boutle, I., Smith, D., Bergot, T., Lac, C., Duconge, L., McGregor, J., Kerr-Munslow, A., Pickering, M., and Clark, R.: LANFEX: a field and modelling study to improve our understanding and forecasting of radiation fog, B. Am. Meteorol. Soc.,* [https://doi.org/10.1175/BAMS-D-16-0299.1](https://doi.org/10.1175/BAMS-D-16-0299.1)*, 2018.* [a](), [b](), [c](), [d]()

-*Zhao, L., Niu, S., Zhang, Y., & Xu, F. (2013). Microphysical characteristics of sea fog over the east coast of Leizhou Peninsula, China. Advances in Atmospheric Sciences, 30, 1154-1172.*